# Evaluation of Bacterial Population Changes and Ecological Dynamics in Oil-Impacted Soils Using 16S rRNA Amplicon Sequencing

**DOI:** 10.3390/biology14081074

**Published:** 2025-08-18

**Authors:** Teddie O. Rahube, Ramganesh Selvarajan, Batendi Nduna, Bokani Nthaba, Loago Molwalefhe, Elisha Shemang

**Affiliations:** 1Department of Biological Sciences and Biotechnology, School of Life Sciences, Botswana International University of Science and Technology, Private Bag 16, Palapye 10071, Botswana; ndunab@biust.ac.bw; 2Department of Environmental Sciences, University of South Africa (UNISA)–Florida Campus, Roodepoort 1702, South Africa; ramganesh.presidency@gmail.com; 3Department of Biochemistry, J.J. College of Arts and Science (Autonomous), Sivapuram, Pudukkottai 622422, India; 4Department of Earth and Environmental Science, School of Earth Sciences and Engineering, Botswana International University of Science and Technology, Private Bag 16, Palapye 10071, Botswana; nthabab@biust.ac.bw (B.N.); molwalefhel@biust.ac.bw (L.M.)

**Keywords:** used motor oil, soil contamination, 16S rRNA amplicon sequencing, metagenomics, bacterial diversity, soil health

## Abstract

Used motor oil is a common pollutant that can harm soil health by altering its microbial communities. In this study, we examined how bacterial diversity and composition changed in soils contaminated with used motor oil at different depths. Using 16S rRNA gene sequencing, we found that oil contamination significantly shifted the microbial community from being dominated by Gram-negative bacteria (such as *Proteobacteria*) in control soils to Gram-positive bacteria (such as *Firmicutes*) in contaminated soils. Statistical analysis (PERMANOVA) confirmed that bacterial communities were significantly different between control and oil-treated soils (Pseudo-F = 3.14, R^2^ = 0.825, *p* = 0.042). Some bacteria potentially involved in breaking down hydrocarbons, such as *Aerococcus*, became more abundant, while beneficial plant-associated bacteria like *Methylobacterium* and *Bradyrhizobium* declined. These findings improve our understanding of how oil pollution affects soil microbes and can guide strategies for restoring contaminated soils.

## 1. Introduction

Soils are vibrant and diverse natural entities situated at the juncture between Earth, air, water, and life. They are essential providers of ecosystem services that support human survival [1]. Soil sustains ecosystems by serving as a host and habitat for diverse organisms, including plants, animals, and microbes [2]. The typical composition of soil includes five primary components: organic matter, minerals, gases, liquids, and microorganisms. These components interact and work together to support vital functions that directly or indirectly affect the health of humans, animals, and the environment. Soil pollution typically involves the introduction of man-made, foreign chemicals that subsequently alter soil properties. Such alterations may lead to undesirable ecological consequences. International organizations widely acknowledge soil pollution as a significant menace to soil health, causing land degradation and a decline in both terrestrial and aquatic biodiversity [3]. Activities that result in soil pollution are largely anthropogenic, such as improper waste disposal or runoff of residues. These residues can include fertilizers, pesticides, antibiotics, heavy metals, oil, and petroleum, primarily originating from various sectors such as agriculture, pharmaceuticals, mining, textiles, and the automobile industry [4,5,6].

Used motor oil (UMO) represents one of the numerous improperly disposed chemicals, leading to environmental pollution, particularly in developing countries. UMO carries a load of polycyclic aromatic hydrocarbons (PAHs) due to the incomplete combustion of fuel, which classifies it as hazardous waste with the potential harm to humans, animals, and the environment [7,8]. In Botswana, a considerable number of unlicensed auto-mechanical repair shops are in residential areas. Regrettably, no research has been conducted to discern the extent and implications of UMO pollution in the country. This scientific data deficit leads to the unavailability of evidence-based guidance for environmental protection policies. Consequently, due to this knowledge gap, Botswana faces a heightened risk of damaging its natural biodiversity. Such a situation could have global health ramifications associated with ecological issues such as climate change. Microorganisms such as bacteria, fungi, archaea, and protozoa are essential components of soil composition, playing a key role in biogeochemical cycling and the degradation of xenobiotic compounds [9]. Among these microorganisms, bacteria are the most abundant in soil, estimated at up to ten billion cells per gram of soil [10]. Despite this abundance, it is believed that less than two percent of these bacteria can be studied through traditional culture methods [11]. The introduction of the polymerase chain reaction (PCR) has significantly advanced the detection and quantification of bacterial genes, leading to improvements in DNA-based molecular approaches. Current research increasingly utilizes next-generation sequencing (NGS) techniques, such as amplicon sequencing and metagenomics, to profile bacterial communities and assess their abundance, diversity, and functions across various uncultured samples. Metagenomics NGS involves high-throughput sequencing platforms like Illumina, PacBio, and Oxford Nanopore to generate vast amounts of unbiased data from complex mixtures of microbial communities in uncultured samples. These metagenomics approaches, including Shotgun and amplicon-based metagenomics, have made substantial contributions to microbial ecology research and have been employed in diverse ecosystems, including contaminated water, soil, and air [12,13,14].

Interestingly, 16S rRNA gene-based metagenomics has also been effectively used worldwide, including in developing countries like Botswana, Africa, to profile community structures and predict the functional potential of microbes in extreme environments [15,16]. Linked to the current study, an investigation was previously carried out to monitor hydrocarbon contamination at an oil contamination experimental site at Botswana International Science of Technology (BIUST) in Botswana, using electrical resistivity imaging (ERI) [17]. The previous study focused on demonstrating the utility of time-related 3D ERI methods to monitor progressive changes in soil composition following an oil spill. Despite this focus, fundamental questions persisted regarding the relationship between UMO contamination and the adaptability and biodegradation capabilities of indigenous microorganisms. Addressing this knowledge gap is key, as comprehending how indigenous microorganisms react to and interact with hydrocarbon contamination is crucial for developing effective remediation strategies.

This study seeks to showcase the effectiveness of employing the 16S rRNA gene metagenomics approach to profile the bacterial community in the soil before and after treatment with UMO. It examines the fluctuations in bacterial relative abundance, species richness, and biodiversity to shed light on the impact of UMO on the soil ecosystem. Moreover, it provides insights into the consequences of the reduction or enhancement of specific bacterial taxa, spanning from phylum to genus levels, and their respective roles in soil ecosystem health and bioremediation activities.

## 2. Materials and Methods

### 2.1. Site Description and Experimental Design

The climate in Botswana is semi-arid; almost throughout the country, the weather is hot and dry with unpredictable rains during the summer months (November–April). This study was carried out on the campus of BIUST, located in Palapye, eastern Botswana, Africa. The BIUST campus is a newly established university site covering 2500 hectares of land and has no documented history of anthropogenic influences. The experimental site is located at latitude 22.59460 S and longitude 27.12330 E, as previously described by Nthaba et al. [17] (Appendix A). The site is characterized by clays and clay-dominated soils displaying minimal grain size variation to a maximum excavated depth, which tends to delay the percolation of pollutants in the subsurface. A pit of 2 m by 4 m was excavated to loosen the soil and prepare an impervious base. A plastic membrane was used to seal the base and the lower walls of the pit to confine the motor oil migration within the walls of the pit and to a maximum depth of 2 m. The pit was then refilled with the same excavated material to about 0.3 m beneath the ground level, and approximately 30 L of used motor oil (UMO) was subsequently spilled evenly on the 0.3 m depth surface. This volume was selected to simulate a moderate, localized contamination event, representative of accidental spills and/or informal dumping often observed in semi-urban and rural environments. The UMO was then entirely covered with the remaining excavated material to the ground level. Contaminants are usually disposed of directly in the soil, posing a high risk to the environment. Impermeability of the pit base and the lower walls is paramount for avoiding the unplanned leakage of hydrocarbon contaminants to the subsurface with the potential to pollute other ecosystems, including groundwater. The dissolution of contaminants such as UMO into the groundwater greatly affects groundwater quality [18] Dissolution is one of the fundamental mass transfer processes that occurs when oil is spilled on water. Therefore, we loosened the soil and irrigated the site regularly to simulate an actual case study.

### 2.2. Soil Sample Collection, DNA Extraction, and 16S rRNA Amplicon Sequencing

Before the pit was excavated for the experiment, control (untreated) soil samples were collected. The experiment incorporated three control samples collected in 2017 and a total of eight treatment samples collected in 2017 and 2019. We acknowledge that this time gap may introduce some degree of environmental variability. However, the experimental design aimed to reflect realistic temporal dynamics following oil contamination under field conditions. The control samples were obtained at a depth of 0–10 cm (C1), 42–48 cm (C2), and 70–80 cm (C4); after treatment with UMO, treatment samples were collected at a depth of 0–10 cm above the oil zone (T1A, T1B), 42–48 cm above the oil zone (T2A, T2B), 61–67 cm around the oil zone (T3A, T3B) and 70–80 cm below the oil zone (T4A, T4B). Standard augers were used to collect the soil samples, which were then placed in sterile zip-lock bags. These samples were transported to the laboratory in a cooler box with an ice pack and subsequently stored at −20 °C until DNA extraction was conducted.

Total DNA was extracted in triplicate from a homogeneous soil sample (1 g), using the ZR Microbe DNA Extraction Kit from Zymo Research Orange, CA, USA, following the manufacturer’s guidelines. These technical replicates were pooled to generate a single composite DNA sample per treatment and depth, which was then used for downstream analysis. The pooled DNA was quantified and assessed for purity using a NanoDrop spectrophotometer (Lasec, Jenway Genova Nano Cape Town, WC, South Africa) at an absorbance of 260 nm. After quantification, all DNA samples were stored at −20 °C until further processing. The DNA samples were then processed for 16S rRNA gene amplicon sequencing on the Illumina MiSeq system, following a bacterial metagenomics workflow as described by Klindworth et al. [19]. Briefly, the pooled genomic DNA samples were amplified through PCR using a universal primer pair, 341F and 785R, which target the V3 and V4 region of the 16S rRNA gene. The resulting amplicons were purified using gel electrophoresis, and Illumina-specific adapter sequences were ligated to each amplicon. After library quantification and individual indexing of the samples, a further purification step was undertaken. The amplicons were sequenced using a 600-cycle MiSeq v3 kit (Illumina Inc San Diego, CA, USA). Each sample generated 20 Mb of data in the form of 2 × 300 bp paired-end reads.

### 2.3. Bioinformatics and Statistical Analyses

Before the bioinformatic analysis, the quality of the raw sequences obtained from the sequencing process was assessed using FastQC to ensure data integrity. Subsequently, the raw sequences were processed using the QIIME2 pipeline (Quantitative Insights into Microbial Ecology, v1.8.0 qiime.org) [20]. This included adaptor and primer sequence removal and quality filtering, trimming, denoising, and merging using the DADA2 algorithm [21] to infer amplicon sequence variants (ASVs), which represent biologically relevant variants. Taxonomic classification of the ASVs was accomplished by using the bacterial sequences database—Silva v138 (https://www.arb-silva.de/, accessed on 15 July, 2025). A taxa filter was implemented to remove sequences originating from other eukaryotic organisms, chloroplasts, and mitochondria. Various metrics were calculated to assess the diversity and richness of the bacterial communities, including the richness estimator (Chao1), diversity indices (Shannon and Simpson), and Goods coverage. The resulting data were utilized to generate bar plots and heat maps representing taxonomic composition at different levels using Origin Pro 22 software. To identify significant differences in the bacterial community between control and treatment groups, the Mann–Whitney U test was applied (*p* < 0.05). The network was constructed using Spearman rank correlation coefficients to assess pairwise relationships in relative abundance among genera. Only statistically significant correlations were included, with a threshold of |ρ| ≥ 0.6 and a *p*-value < 0.01, adjusted for multiple comparisons using the Benjamini–Hochberg false discovery rate (FDR) correction to control for Type I errors. Further, to assess microbial differences between oil-impacted treatments and controls, differential abundance analysis was performed at the genus level using DESeq2. Low-abundance taxa (total abundance ≤ 10) were filtered out, and the resulting phyloseq object was converted to a DESeq2 object with size factors estimated using the “poscounts” method. Comparisons were made between treatments and controls using default settings, and significantly altered taxa were identified based on an adjusted *p*-value < 0.05. Enhanced volcano plots were generated using the “EnhancedVolcano, v1.27” R package to visualize significant taxonomic shifts.

## 3. Results

### 3.1. Bacterial Diversity Analysis

After the completion of the filtering, denoising, and removal of chimeric sequences, the final sequences exhibited a range from 33,374 (T3B) to 119,809 (T1A). The percentage of sequences that remained non-chimeric, relative to the original input, varied among the samples, spanning from 64.54% (T4A) to 80.57% (C2). In terms of non-chimeric sequence percentages, the control samples, specifically C1, C2, and C4, displayed values of 68.98%, 80.57%, and 78.3%, respectively. In contrast, the treatment groups exhibited the following percentages: 73.58% (T1A), 65.05% (T1B), 66.62% (T2A), 68.37% (T2B), 67.82% (T3A), 69.72% (T3B), 64.54% (T4A), and 66.44% (T4B). The alpha diversity of the samples, characterized by multiple indices, displayed significant changes across different soil depths and between the control and treatment groups. At a depth of 10 cm, Control 1 (C1) demonstrated the highest diversity with 1264 ASVs, a Chao1 index of 1278, a Shannon index of 9.39, a Simpson index of 0.997, and an evenness index of 0.911. In contrast, Treatment 1 samples (T1A and T1B) exhibited lower diversities after the treatment process. At 48 cm, Control 2 (C2) displayed lower diversity indices compared to Treatment 2 samples (T2A and T2B), with T2B registering the highest values. For the samples collected at 65 cm, both T3A and T3B exhibited similar levels of diversity. Lastly, at a depth of 80 cm, Control 4 (C4) had the lowest diversity among all the samples, while Treatment 4 samples (T4A and T4B) demonstrated higher diversity levels in comparison to their respective control samples (Figure 1). The PCoA plot, derived from Bray–Curtis dissimilarity values between microbial community profiles, reveals clear clustering patterns among the samples (Appendix A). Control samples (C1–C3) group distinctly from treatment samples (T1A–T4B), indicating substantial differences in microbial composition. These patterns were supported by permutational multivariate analysis of variance (PERMANOVA), which detected significant differences among groups (Pseudo-F = 3.14, R^2^ = 0.825, *p* = 0.042, 999 permutations). The high R^2^ value indicates that group identity explained approximately 82.5% of the variation in community structure.

### 3.2. Bacterial Composition at Phylum Level

The analysis of the bacterial composition in different samples revealed statistically significant differences among the treatments and controls (*p* < 0.05). Control 1 (C1) exhibited a significantly higher relative abundance of *Actinobacteria* compared to other samples, making up 54.3% of the total bacterial composition. Conversely, Control 2 (C2) and Control 4 (C4) were significantly different, with a dominance of *Proteobacteria*, constituting 71.5% and 80.6% of their respective bacterial populations (*p* < 0.05). Treatment 1A (T1A) also showed a significant presence of *Proteobacteria*, though at a relatively lower prevalence of 44.6% (*p* < 0.05). Furthermore, the *Firmicutes* phylum was revealed as the predominant bacterial group in Treatment samples, including 1B (T1B), 2A (T2A), 2B (T2B), 3A (T3A), 3B (T3B), 4A (T4A), and 4B (T4B), representing 70.8% to 78.7% of the bacterial communities under the treatment conditions (*p* < 0.05). In terms of other phyla, significant variations were observed, with *Chloroflexi* ranging from 0.07% to 12.11%, *Planctomycetota* from 0% to 11.33%, *Acidobacteriota* from 0.29% to 5.10%, *Gemmatimonadota* from 0.02% to 9.57%, *Verrucomicrobiota* from 0% to 1.31%, and *Myxococcota* from 0% to 0.65% (*p* < 0.05). The variations in Firmicutes from 13.34% to 76.97% were also statistically significant, as were the fluctuations in *Bacteroidota* from 0.27% to 8.75% (*p* < 0.05). Additionally, the collective group of Minor Phyla represented significant proportions ranging from 0.74% to 2.17% of the microbial populations across the samples (Figure 2).

### 3.3. Bacterial Composition at Class Level

The bacterial class distribution also demonstrated significant differences across the control and treatment samples (Figure 3). For instance, C1 showed a significantly higher proportion of *Acidimicrobiia* (7.18%), *Actinobacteria* (12.85%), and Actinobacteriota (1.55%) (*p* < 0.05). A shift in dominance was observed in C2, with Alphaproteobacteria taking the lead (40.20%), followed by Bacilli (14.72%), and a reduced presence of *Actinobacteria* and *Acidimicrobiia* at 3.44% and 0.53%, respectively. Furthermore, there is clear dominance of Alphaproteobacteria (44.99%) and Gammaproteobacteria (35.60%) in the C3 sample (*p* < 0.05). Sample T1A exhibited a high prevalence of Alphaproteobacteria (25.19%), accompanied by a notable presence of Bacilli (7.73%) and Gammaproteobacteria (19.37%). Similarly, the prevalence of Clostridia (51.59%) and Bacilli (24.40%) in T1B, and the dominance of Clostridia (51.87% and 47.41%) and Bacilli (24.69% and 23.38%) in T2A and T2B, respectively, were all significant (*p* < 0.05). The samples T3A and T3B maintained the trend of Clostridia and Bacilli dominance, constituting 49.80% and 51.87%, as well as 24.77% and 25.21% of the respective communities. Finally, T4A and T4B continued with this pattern, harboring most Clostridia (52.07% and 52.84%) and Bacilli (24.84% and 25.80%).

### 3.4. Bacterial Abundance at Genus Level

A detailed examination of the bacterial genus distribution (Figure 4) across different sample groups revealed the *Methylorubrum* genus to be ubiquitous, with substantial variations in its relative abundance. While it achieved peak prevalence in the Treatment 2A group at 24.47%, it displayed relatively low concentrations in the other groups, varying from 2.15% to 2.44%. This noticeable variation suggests the potential selectivity of Treatment 2A in promoting the proliferation of *Methylorubrum*, a trait absent in the other treatments and control groups. *Escherichia-Shigella* and *Bradyrhizobium* genera were also present in significant proportions across the different treatments, again with evident variation. The *Comamonadaceae* genus showed an interesting pattern, with a significant rise in its relative abundance across all treatment groups, surpassing 2%. Its maximum relative abundance, however, was found in the Control 4 group at 7.91%. This might imply that the conditions in the Control 4 group are especially favorable to the proliferation of this bacterial genus. Meanwhile, *Bacillus* and *Paenibacillus* genera showed relatively limited occurrence, suggesting that these genera were less influenced by the applied treatments. In Treatment 3, notable abundance was observed for the *Lactobacillus* and *Clostridium_sensu_stricto_1* genera. The *Lactobacillus* genus exhibited an evident increase in Treatment 3A and 3B, peaking at 3.91% and 3.42%, respectively. In contrast, *Clostridium_sensu_stricto_1* exhibited a markedly higher abundance, reaching 28.58% in Treatment 3A and 25.18% in Treatment 3B. This substantial increase suggests a selective stimulatory effect of Treatments 3A and 3B on *Clostridium_sensu_stricto_1*. A similar trend was observed for *Romboutsia*, which showed high prevalence exclusively in the Treatment 3A and 3B groups. While these shifts suggest that certain conditions in Treatment 3 may influence the enrichment of these genera, no direct functional role in hydrocarbon degradation is implied without further experimental validation. Functional inference would require complementary metagenomic or culture-based analyses to confirm potential hydrocarbon-degrading capabilities.

### 3.5. Bacterial Variation Between Control and Treatment

The Mann–Whitney U test revealed statistically significant differences (*p* < 0.05) in bacterial community composition between specific control and treatment pairs. Notably, a significant change was observed between Control 1 and Treatment 1, while other control–treatment pairs did not show any significant differences. In terms of bacterial composition, Treatment 3 exhibited differentiation from Treatments 1 and 2 but not from Treatment 4 (Figure 5). A closer look at genus-level changes revealed distinct patterns. In Treatment 1, a sharp decline in g_67-14 (class *Thermoleophilia*) and uncultured taxa was observed compared to Control 1, while g_*Methylobacterium-Methylorubrum* and g_*Escherichia-Shigella* showed marked increases, suggesting a strong stimulatory effect of Treatment 1. In contrast, Treatment 2 exhibited significant reductions in *g_Methylobacterium-Methylorubrum*, *g_Escherichia-Shigella*, *g_Bradyrhizobium*, *g_Ralstonia*, and *g_Bacillus*, relative to Control 2. However, *g_Rubrobacter* and *g_Lactobacillus* appeared relatively enriched under this condition. For Treatment 4, similar declines were observed in *g_Methylobacterium-Methylorubrum*, *g_Escherichia-Shigella*, and *g_Bradyrhizobium* compared to Control 4, whereas *g_Bacillus* and *g_Lactobacillus* showed elevated counts, indicating a potential adaptation or resistance under oil-exposed conditions. Treatment 3 presented a unique microbial profile, with significantly higher counts of *g_Clostridium_sensu_stricto_1* and *g_Romboutsia* (class *Clostridia*), which were far more abundant compared to Treatments 1 and 2. Additionally, *g_Turicibacter*, *g_Muribaculaceae*, and *g_Lactobacillus* were also notably enriched. Interestingly, while *g_Methylobacterium-Methylorubrum* declined in most treatments, it showed a mild resurgence in Treatment 3, indicating that oil exposure and experimental conditions exert selective pressures that shape bacterial community structures. Although *Methylobacterium* is often associated with plant health and nitrogen fixation, our data do not include direct functional measurements to confirm the ecological impact of its decline. As such, any implications for soil health remain speculative. Without experimental validation such as plant–soil microcosm assays or nitrogen fixation rate measurements, we cannot establish a causal relationship between the observed taxonomic change and ecosystem function.

### 3.6. Co-Occurrence Network Analysis at the Bacterial Genus Level

The co-network analysis of the top 30 bacterial genera across various control and experimental conditions provides some valuable insight into the complex interactions within the microbial community (Figure 6). The resulting network included both positive and negative correlations, highlighting the dynamics of potential microbial cooperation or competition. For instance, the genus *Vicinamibacteraceae* demonstrates a notable ability to maintain positive relationships across multiple conditions, with a significant positive correlation indicating strong associations with other genera. In contrast, g_67 exhibited a dominance of negative correlations, particularly under experimental conditions, indicating possible niche competition or antagonistic interactions in response to UMO contamination. Genera like *Methylobacterium* and *Escherichia* show a network of positive correlations that are particularly strong in experimental conditions, implying that these genera may thrive or respond similarly under the conditions tested in the experiments. *Romboutsia* and *Turicibacter* have a combination of positive and negative correlations across both control and experimental conditions. Notably, uncultured, *Burkholderia*, and *Providencia* are characterized by several significant positive correlations that are higher in experimental conditions compared to control, suggesting that the experimental manipulations may favor their association with other genera. The absence of significant negative correlations for genera such as *Rubrobacter* and *Alcaligenes*, and for several uncultured genera, across both control and experimental conditions, is indicative of either a non-competitive stance or a broad ecological niche that allows for coexistence without direct antagonism. while others exhibit a balance of positive and negative correlations across control and experimental conditions. This pattern suggests that experimental manipulations can either promote cooperation or competition among genera, highlighting the dynamic nature of microbial interactions in response to environmental changes.

### 3.7. Differential Abundance Analysis

The analysis of bacterial population changes in oil-impacted soils revealed notable microbial shifts across different treatments compared to the control. Control vs. Treatment 1 showed significant changes in *Ruminococcus_gauvreauii_group*, *Aerococcus*, *AKYG1722*, *Clostridium_sensu_stricto_1*, *Thermoleophilia*, *Prevotellaceae*, *Dubosiella*, *Facklamia*, *Faecalibaculum*, *JCM_18997*, *Lactococcus*, *Muribaculaceae*, *Romboutsia*, *Turicibacter*, and uncultured4, indicating microbial restructuring likely associated with contaminant degradation (Figure 7). Control vs. Treatment 2 exhibited shifts in *Ruminococcus_gauvreauii_group*, *Aerococcus*, *Clostridium_sensu_stricto_1*, *Prevotellaceae*, *Dubosiella*, *Facklamia*, *Faecalibaculum*, *Lactococcus*, *Muribaculaceae*, *Romboutsia*, and *Turicibacter*, suggesting microbial responses that may facilitate soil recovery. Similarly, Control vs. Treatment 3 showed significant changes in *Ruminococcus_gauvreauii_group*, *Aerococcus*, *Clostridium_sensu_stricto_1*, *Prevotellaceae*, *Dubosiella*, *Facklamia*, *Faecalibaculum*, *Lactococcus*, *Muribaculaceae*, *Romboutsia*, and *Turicibacter*, indicating microbial competition, adaptation, or bioremediation potential. Control vs. Treatment 4 displayed shifts in *Ruminococcus_gauvreauii*_group, *Aerococcus*, *Clostridium_sensu_stricto_1*, *Prevotellaceae*, *Dubosiella*, *Facklamia*, *Faecalibaculum*, *Lactococcus*, *Muribaculaceae*, *Romboutsia*, *Turicibacter*, and uncultured13, further supporting microbial adaptation and potential bioremediation activity. While certain bacterial taxa exhibited consistent changes across all treatments, highlighting their key role in oil degradation, unique microbial responses in individual treatments suggest treatment-specific ecological effects.

## 4. Discussion

Soil bacterial diversity is known to be influenced by changes in the environmental physicochemical factors such as temperature, pH, water content, nutrients, texture, and vegetation types [16,22]. In this study, we observed high diversity and variation both within and across different treatment groups and depths, highlighting the complexity of the soil samples. At 0–10 cm, which is characteristic of topsoil, there is high variation compared to other depths. Considering T1A and T1B, topsoil collected in 2017 and 2019, respectively, it is not surprising, considering the direct exposure of topsoil to environmental factors, especially temperature. The influence of broader factors such as climate change and human activities cannot be overlooked. Rising temperatures and increased frequency of extreme weather events can alter soil conditions, affecting microbial life [23]. While our dataset includes samples collected in 2017 and 2019, the two-year gap between time points prevents a continuous assessment of microbial succession. As such, our findings should be interpreted as comparative snapshots of community structure at two stages rather than a detailed temporal trajectory.

In this study, a total of 10 major phyla and 22 major classes were observed in the topsoil of the control samples. Considering only the subsurface (due to expected variation in the topsoil), there is a significant shift in bacterial community composition between control and treatment in the corresponding soil depth. The phylum *Proteobacteria*, which is most predominant in control samples when compared to treatment samples, is replaced by the phylum Firmicutes, which appears to be predominant in the treatment samples. The phylum Proteobacteria, known for its diverse metabolic capabilities and predominance in various environments, is commonly affected by soil disturbances, including anthropogenic activities. In contrast, Firmicutes, with many members known for their resilience and ability to degrade hydrocarbons, often increase in relative abundance in contaminated sites. This pattern has been observed in other studies, which reported an increase in Firmicutes in oil-contaminated soils, reflecting their potential role in hydrocarbon degradation [24,25,26]. Proteobacteria is a phylum characterized mainly by Gram-negative bacteria, while Firmicutes are characterized mainly by Gram-positive bacteria. The decrease in Gram-negative bacteria and the enrichment of Gram-positive bacteria in contaminated soils reveal that hydrocarbon contamination can selectively inhibit certain microbial groups while favoring others, particularly those capable of hydrocarbon degradation. Further, the shift in microbial communities due to contamination not only reflects the resilience and adaptability of soil bacteria but also has implications for soil health. As Sharma et al. pointed out, changes in microbial community composition can affect soil fertility, structure, and its capacity to support plant life [27].

### 4.1. Enrichment of Bacterial Genera: Implications for Soil Ecosystem Function and Health

The bacterial genera *Muribaculaceae*, *Prevotella*, *Aerococcus*, *Romboutsia*, *clostridium_sensu_stricto*, *Dubosiella*, *Faecalibaculum*, and *Turicibacter*, all categorically linked to the major phylum *Firmicutes*, were enriched following exposure to UMO in this experiment. This taxonomic enrichment may reflect adaptive microbial strategies in response to hydrocarbon pollutants, but it also warrants deeper investigation into the metabolic pathways and functional genes that facilitate this enrichment under such selective pressures. Of the enriched genera, only *Prevotella* and *Aerococcus* could be directly associated with positive ecosystem services such as biodegradation of xenobiotic and recalcitrant pollutants. *Prevotella*’s primary involvement in crude oil degradation was first reported by Mukjang et al. [28], with functional predictions indicating the presence of genes encoding for alkane hydroxylases, enzymes essential for initiating alkane degradation. *Aerococcus*, likewise, has been identified in hydrocarbon-contaminated environments [29,30] and is suspected to contribute to aerobic degradation of polycyclic aromatic hydrocarbons (PAHs) via enzymes such as monooxygenases and dioxygenases. Interestingly, many bacterial genera of animal origin, associated with human and animal gut microbiota, were found to have increased in relative abundance. For instance, *Turicibacter*, often found in compost and enriched in anaerobic digestion systems, is known to harbor carbohydrate-active enzymes (CAZymes) involved in lignocellulose degradation [31]. *Romboutsia*, a genus within the *Peptostreptococcaceae* family within the *Firmicutes*, also demonstrates genomic potential for fermentative carbohydrate metabolism, including genes linked to glycolysis, butyrate synthesis, and other short-chain fatty acid (SCFA) production pathways [32]. These functional attributes may explain their enrichment under nutrient-rich but stressed conditions, as seen in hydrocarbon-impacted soils. Co-occurrence network analysis revealed complex interactions among *Romboutsia* and *Turicibacter*, possibly reflecting syntrophic relationships or overlapping functional niches related to substrate degradation and stress tolerance. Similarly, *Dubosiella*, *Faecalibaculum*, and *Muribaculaceae* exhibit capabilities for degrading complex polysaccharides, and their potential involvement in anaerobic fermentation processes in oil-contaminated soils merits further exploration [33,34].

*Clostridium_sensu_stricto*, considered a true genus of *Clostridium* [35], is known for its anaerobic fermentative metabolism, including the production of volatile fatty acids (VFAs) and hydrogen gas. Some species have demonstrated activity in the decomposition of *Microcystis* biomass [36], and their ability to form resilient endospores enhances their survival under harsh environmental conditions, such as those created by hydrocarbon contamination [37]. A substantial proportion of unclassified bacteria also appears in higher abundance in the treatments relative to controls. These unidentified and mysterious representatives warrant further investigation; these unclassified populations may have both positive and negative impacts on soil ecosystem function and health. Our study further emphasizes and demonstrates the power of utilizing a 16S rRNA gene-based metagenomics approach towards uncovering novel microbial diversity, even while acknowledging the limitations in functional prediction. However, recent advances in shotgun metagenomics and functional annotation tools offer hope for the more robust prediction of the functional potential of uncultured and/or unclassified taxa, particularly with regard to key pathways like alkane degradation (alkB), aromatic ring cleavage (catA, pcaH), and biosurfactant synthesis [38].

### 4.2. The Loss of Bacterial Genera and Its Impact on Soil Ecological Functions

The family *Vicinamibacteraceae*, the first described within *Acidobacteria*, constitutes a globally widespread group of Gram-negative, non-spore-forming, aerobic, chemo-organoheterotrophic bacteria inhabiting soil environments [39]. As observed in the co-network analysis, *Vicinamibacteraceae* may play a key role in the microbial ecosystem across varied environmental states, as revealed in both control and experimental settings. *Vicinamibacteraceae* are known to be functionally versatile and play many important roles, including nutrient mobilization during decomposition, thus maintaining soil ecosystem health [40]. When disregarding the topsoil, there is some light evidence suggesting the loss of certain bacterial genera within the phylum *Proteobacteria* following soil treatment with UMO. Notable bacterial genera representatives of the phylum *Proteobacteria*, *Methylobacterium-Methylorumbrum*, *Escherichia-Shigella*, *Bradyrhizobium*, *Ralstonia*, and *Phyllobacterium*, significantly declined in relative abundance in the soil subsurface and below. Several ecological mechanisms may underlie this decline. The inherent toxicity of UMO-derived hydrocarbons can adversely affect sensitive microbial taxa by impairing membrane integrity or enzyme activity [41]. Additionally, environmental filtering likely occurs as UMO alters soil physicochemical parameters (e.g., oxygen availability, redox status, nutrient profiles), creating selective conditions that exclude less tolerant species [42] Furthermore, niche exclusion may result from the proliferation of stress-adapted or hydrocarbon-degrading microbes that outcompete native beneficial genera through competition for limited resources [43].

*Methylobacterium* and *Escherichia* show a network of positive correlations that are particularly strong in experimental conditions, implying that these genera may thrive or respond similarly under the conditions tested in the experiments. Genus *Methylobacterium-methylorumbrum* comprises closely related facultatively methylotrophic bacterial species, and some of the literature reported their use in bioaugmentation and efficiency in the biodegradation of polyaromatic hydrocarbons in contaminated environments [44,45]. Certain species of *Methylobacterium* are also known to be of agricultural importance, plant-associated bacteria, and model organisms in microbiology [46], playing a vital role as biostimulators by producing phytohormones and providing important nutrients to plants as they fix nitrogen and solubilize phosphorus and iron to promote plant growth [47]. The biotechnological role of other *Methylorumbrum* species in environmental bioremediation has also been recently documented in the literature [48,49]. Similarly, *Bradyrhizobium*, symbiotic bacteria, encode multiple functions that are critical to plant growth, including nitrogen fixation and nodulation [50]. *Phyllobacterium* is another well-researched plant probiotic that has not been associated with causing diseases in humans and is described as a good candidate for use as a biofertilizer, supplying phosphorus for plants, especially during dry seasons [51]. In contrast, genera like *Escherichia-Shigella* and *Ralstonia* are often associated with negative impacts on human and plant health, suggesting a complex interplay of beneficial and harmful microbial elements within the same ecosystem. This duality emphasizes the need for a better understanding of microbial community compositions, especially in the milieu of environmental changes and soil–ecosystem health.

## 5. Conclusions

This study sheds light on the profound impact of UMO contamination on soil ecosystem health, particularly through its alteration of bacterial diversity and relative abundance. The presence of oil pollution in the soil leads to significant ecosystem alterations, marked by notable changes in bacterial community structure. This includes both the depletion and proliferation of specific bacterial genera, thereby influencing the ecological functions of the soil. This study successfully identified key bacterial genera and their potential roles, providing valuable insights that could inform future strategies for the effective management of soil contamination in similar regions worldwide. Furthermore, the use of the 16S rRNA amplicon sequencing approach was found to be effective in providing high resolution of the ASVs down to the genus level. Consequently, this study encourages the expanded use of the 16S rRNA gene metagenomics approach in long-term environmental pollution studies to fully understand the roles and relationships among the microbial populations in a contaminated environmental setting. However, one limitation of this study is the absence of independent biological and sequencing replicates. Although triplicate DNA extractions were pooled to reduce variability, the lack of replicate sequencing may introduce methodological bias. Future studies should incorporate both biological and technical replicates to improve the robustness and reproducibility of microbial community assessments. Overall, the findings have practical implications for bioremediation and environmental management. Understanding the microbial responses to UMO contamination can guide the development of targeted bioremediation strategies that harness native microbial taxa for pollutant degradation. Moreover, the identified genera can support risk assessment frameworks and inform regulatory policies on soil contamination in similar semi-arid environments.

## Figures and Tables

**Figure 1 biology-14-01074-f001:**
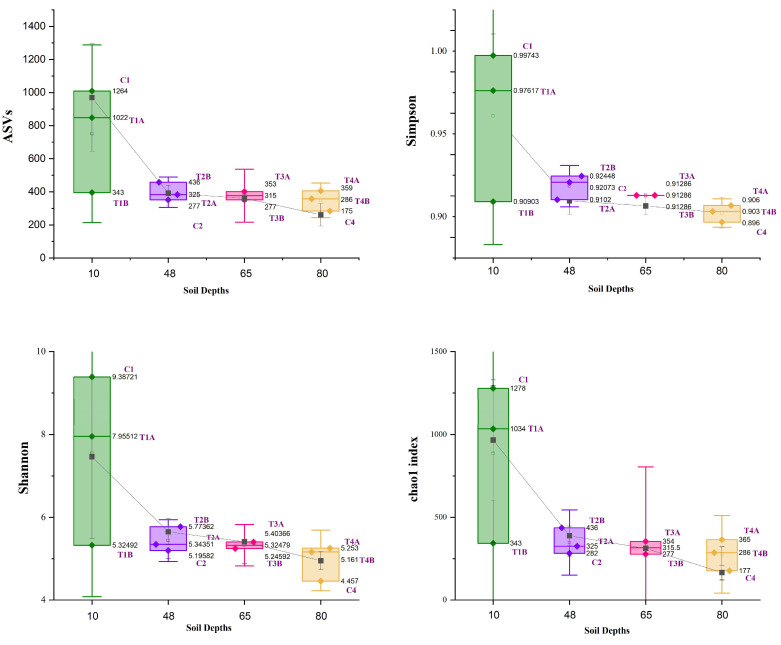
Alpha diversity index of bacterial communities across different soil depths (0–10 cm, 42–48 cm, 61–67 cm, 70–80 cm) and between control (C1, C2, C4) and treatment groups (T1A, T1B, T2A, T2B, T3A, T3B, T4A, T4B).

**Figure 2 biology-14-01074-f002:**
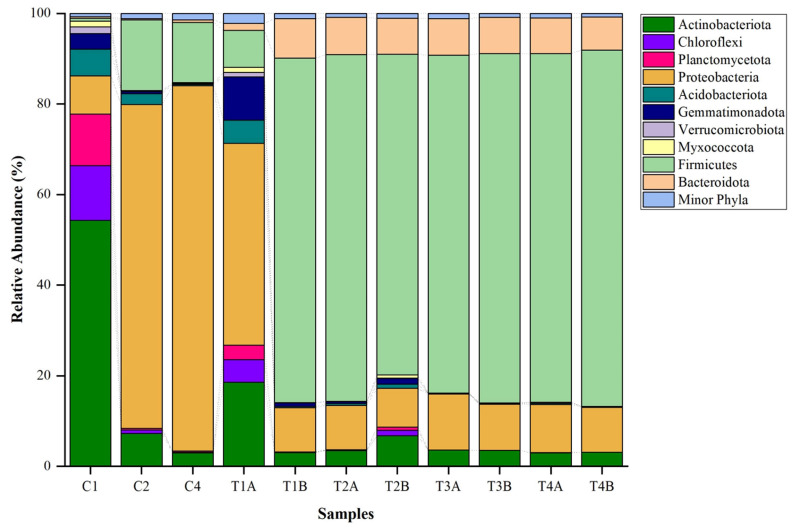
Clustered stacked column chart showing phylum-level distribution of bacterial composition in control (C1, C2, and C4) and treatment soil (T1, T2, T3, T4).

**Figure 3 biology-14-01074-f003:**
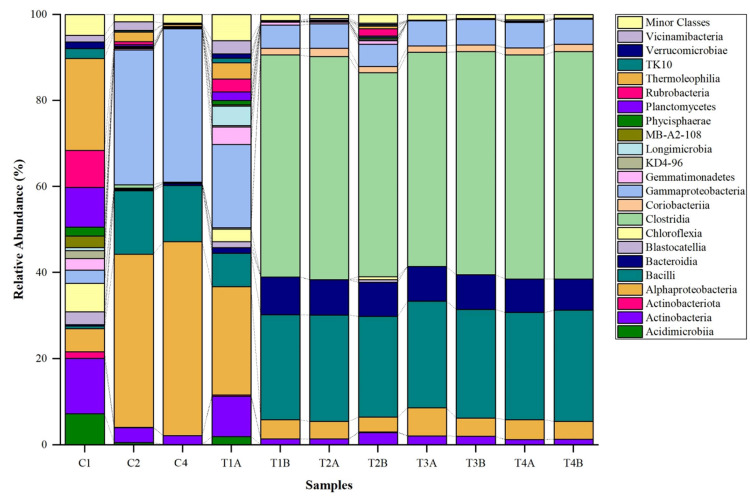
Clustered stacked column chart showing the class-level distribution of bacterial composition in control (C1, C2, and C4) and treatment soil (T1, T2, T3, T4).

**Figure 4 biology-14-01074-f004:**
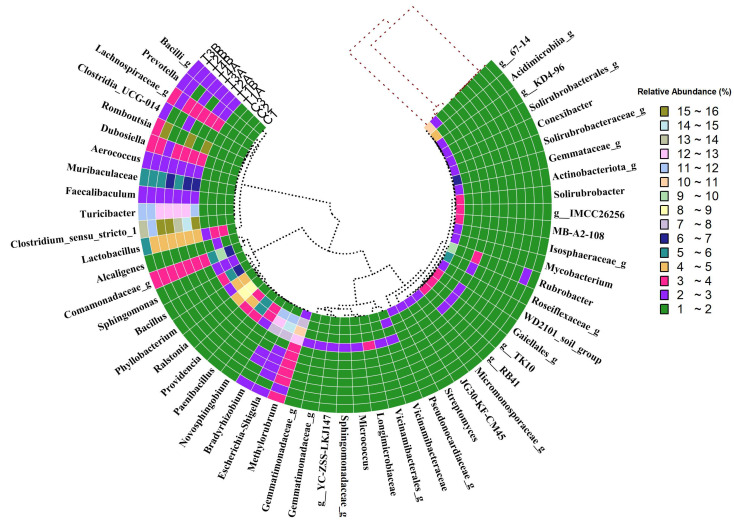
Heat map showing relative abundance of bacterial genera in control (C1–C3) and treatment (T1–T4) soil samples, with color intensity indicating abundance variation.

**Figure 5 biology-14-01074-f005:**
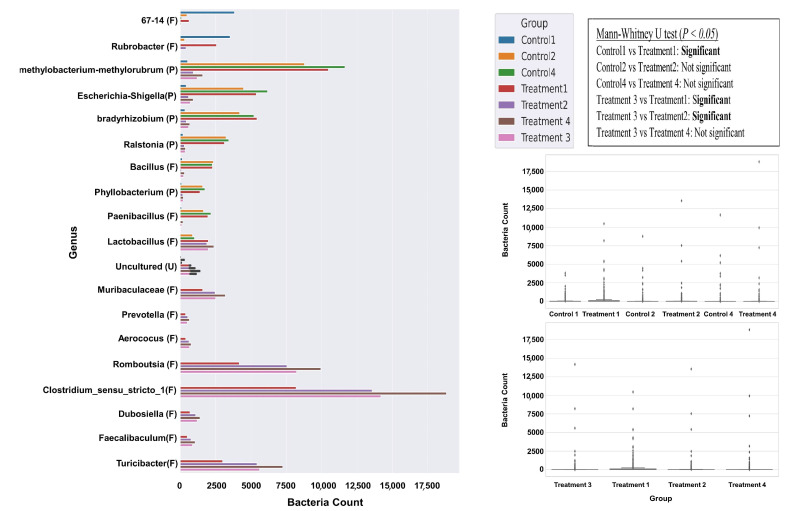
Trends and abundance of the bacterial genus within the major phyla *Firmicutes* (F) and *Proteobacteria* (P) across controls (C1, C2, C4) and treatments (T1, T2, T3, T4). Statistical differences between groups were assessed using the Mann–Whitney U test. Results indicate statistically significant differences (*p* < 0.05).

**Figure 6 biology-14-01074-f006:**
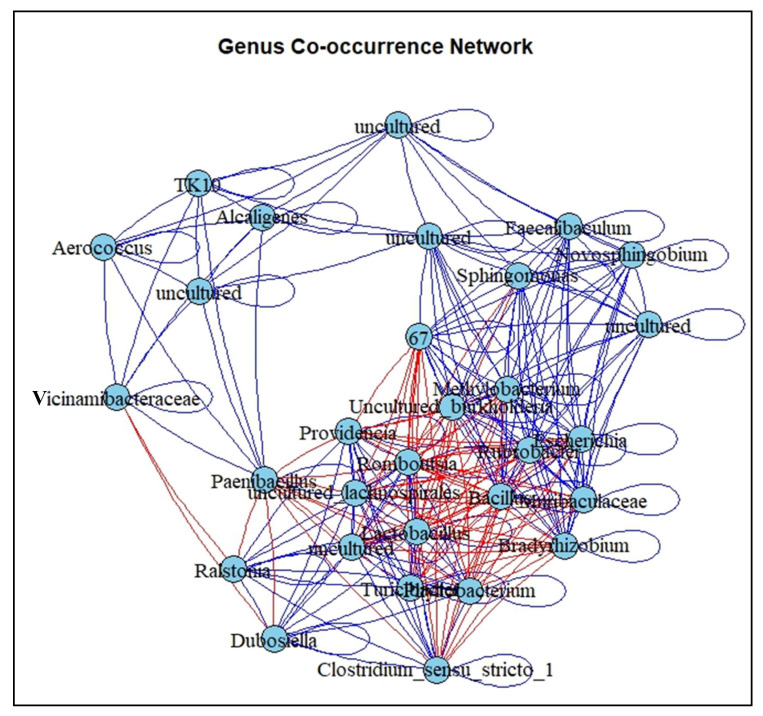
Co-occurrence network analysis at the bacterial genus level. The blue and red edges represent positive and negative correlations, respectively.

**Figure 7 biology-14-01074-f007:**
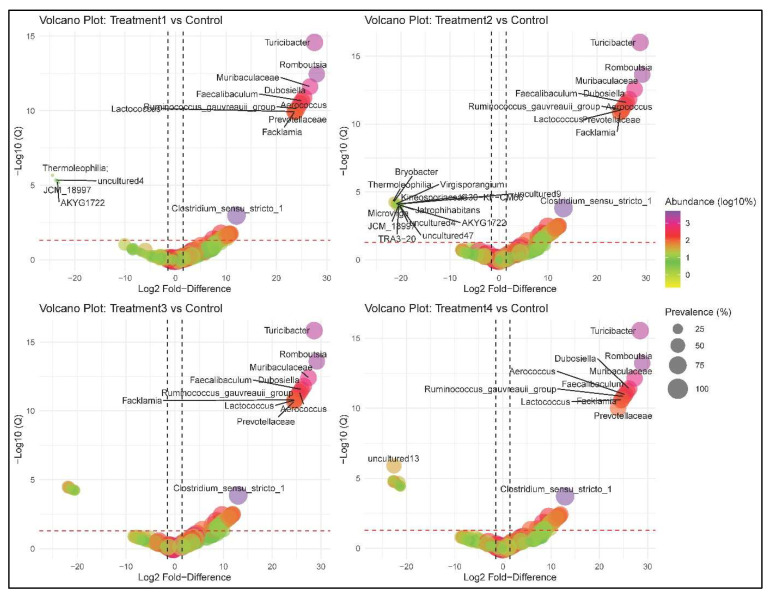
Volcano plots showing bacterial abundance between different treatment groups and control. The *x*-axis represents the Log2 Fold-Difference, while the *y*-axis (−Log10(Q)) shows the statistical significance of these differences.

## Data Availability

The sequence datasets are available at the NCBI SRA under the BioProject Accession PRJNA781210 through the weblink https://www.ncbi.nlm.nih.gov/bioproject/PRJNA781210 (accessed on 15 July 2025).

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
