# Peer review of "Evaluation of Bacterial Population Changes and Ecological Dynamics in Oil-Impacted Soils Using 16S rRNA Amplicon Sequencing"

_biology, 2025, doi:10.3390/biology14081074_

Round 1
Reviewer 1 Report
Comments and Suggestions for Authors
Line 28: consider changing “genus” to “genera”
Line 33: Please use a scientifically more meaningful synonym for the word “stress” for eg, “highlight”.
Lines 45–51: Sentence structure could be simplified for better clarity.
Line 75–76: “Less than two percent...” Please verify this statement with a recent citation. The citation provided mentioned the statistics as a statement without a proper citation.
Please clarify whether this was a replicated experiment or a single-plot (pit) study?
Line 138–140: Kindly explain why there are three control samples versus eight treatment samples. This could affect statistical analysis.
Data regarding the physicochemical parameters of the samples would be a valuable addition.
Line 290-300: Fold change values over −100% or +100% are a bit unusual or misleading; please revalidate or reformat the data presented.
Line 266: "LactoBacillus” should be "Lactobacillus" (genus names must be italicised and properly capitalised). Ensure all genus names are italicised and properly spelt consistently throughout the manuscript.
Line 270: Consider replacing “strongly high representation” with "markedly high abundance" for better scientific clarity.
Line 421: Please change the “bacteria species are adapted..." to “bacterial species”
Comments on the Quality of English LanguageA moderate revision is recommended to improve the overall language quality.
Reviewer 2 Report
Comments and Suggestions for Authors
This study presents a comprehensive analysis of microbial shifts in oil-contaminated soils using 16S rRNA gene amplicon sequencing, comparing control and treated samples across various depths. The experimental design is generally robust, and the results offer valuable insights for informing soil remediation strategies. However, the manuscript would benefit from clarification in the description of methods, refinement of statistical reporting, and more in-depth functional interpretation of key findings. With appropriate revisions, the study could make a meaningful contribution to the field of microbial ecology and environmental biotechnology.
Reviewer Comments
Line 113–135
The experimental design lacks a clear explanation of the rationale for the volume of oil used and its environmental relevance. Was 30 L per pit representative of real-world contamination scenarios?
Line 138–147
The control samples were collected in 2017, but some treatment samples were collected as late as 2019. This time gap may introduce environmental variability. Please explain how this was accounted for in the analysis.
Line 149–162
No mention is made of technical replicates during sequencing. Were the DNA samples from different soil depths sequenced independently or pooled prior to library preparation? This information is critical to assess the reproducibility and robustness of the sequencing data. Without biological or technical replication, the observed shifts in microbial diversity and community composition may be confounded by methodological artifacts or random variability. If replicates were included, please specify their number and how they were incorporated into downstream analyses. If not, the authors should discuss this limitation and interpret their results with appropriate caution. Including technical or biological replicates is a widely accepted best practice in microbiome research, especially when aiming to detect treatment effects or environmental gradients with confidence.
Line 164–179
The manuscript applies the Mann-Whitney U test but does not report p-values or effect sizes for key comparisons. Please include statistical outputs for transparency and reproducibility.
Line 183–201
While alpha diversity indices are well presented, the lack of beta diversity analysis limits understanding of community compositional differences. Consider adding PCoA or NMDS analysis.
Line 208–230
The stacked bar plots at the phylum level are informative, but the addition of a statistical summary table (e.g., ANOVA or Kruskal-Wallis test results) would help validate the observed trends.
Line 251–273
Some genus-level ecological inferences appear speculative without functional validation. For example, assertions about Romboutsia and Turicibacter in hydrocarbon degradation require more cautious interpretation.
Line 283–310
The percentage change values are informative but may be misleading due to low baseline abundance. Consider also reporting absolute abundance differences or normalized read counts.
Line 317–339
The co-occurrence network lacks information on correlation metrics, thresholds, and statistical significance testing. Please specify the methods used to construct and validate the network.
Line 405–444
The discussion focuses on taxonomic shifts but does not integrate pathway-level information (e.g., known hydrocarbon degradation genes). Linking taxonomy with function would strengthen the analysis.
Line 446–481
The observed decline of beneficial genera is important, but the manuscript does not explore potential mechanisms (e.g., toxicity, niche exclusion, environmental filtering) that may explain this pattern.
Line 482–496
The conclusion section should better emphasize the practical implications of the findings, such as how the results inform bioremediation policy or soil contamination management practices.
Reviewer 3 Report
Comments and Suggestions for Authors
The study provides valuable insights into microbial responses to oil contamination but requires stronger mechanistic and ecological validation. The following are my comments and concerns about the weaknesses of this work.
- The study focuses solely on microbial shifts without correlating them with soil physicochemical properties (pH, organic matter, PAH concentrations). Without these data, it is impossible to discern whether microbial changes are driven by oil contamination or other environmental factors (e.g., moisture, nutrient availability). I suggest that authors include soil chemistry data (such as GC-MS for PAHs, elemental analysis) to link microbial shifts directly to oil contamination.
- Samples were collected in 2017 and 2019, but the study lacks a continuous time-series analysis. The two-year gap obscures dynamic microbial responses (early vs. late-stage degraders). The authors are supposed to add intermediate sampling points (such as 3, 6, 12 months) to track succession patterns.
- The study infers biodegradation potential based on taxonomy (such as Aerococcus, Firmicutes) but lacks functional gene analysis (like alkB, nah for hydrocarbon degradation). Taxonomic shifts do not confirm metabolic activity. Firmicutes enrichment could reflect sporulation under stress rather than degradation. The authors should supplement with shotgun metagenomics or qPCR of functional genes.
- Alpha diversity metrics (Chao1, Shannon) are reported but not tested for significance (ANOVA/Kruskal-Wallis). The Mann-Whitney U test is used for pairwise comparisons but may not account for multiple testing (such as FDR correction). So, claims of "significant" shifts may be overstated. Authors need to apply PERMANOVA for beta diversity and correct p-values for multiple comparisons.
- The discussion assumes Methylobacterium decline harms soil health, but no plant/microcosm experiments validate this. Ecological conclusions are speculative without demonstrating functional consequences (reduced nitrogen fixation). Authors are supposed to include microcosm experiments to link microbial shifts to ecosystem functions (such as plant growth assays).
- Samples were taken at 0–80 cm but analyzed as discrete layers (e.g., 0–10 cm, 42–48 cm). Stratification may mask gradients in microbial activity (such as aerobic vs. anaerobic zones). Authors are supposed to analyze continuous depth profiles or finer intervals (for example, every 5 cm).
Round 2
Reviewer 3 Report
Comments and Suggestions for Authors
I acknowledge that the authors have made substantial and thoughtful revisions in response to the previous comments. The manuscript now demonstrates clear improvements in structure, clarity, and scientific rigor. In light of these changes, I recommend that the manuscript be accepted in its current form.